# Physician-Level Cost Control Measures and Regional Variation of Biosimilar Utilization in Germany

**DOI:** 10.3390/ijerph17114113

**Published:** 2020-06-09

**Authors:** Katharina E. Blankart, Friederike Arndt

**Affiliations:** 1CINCH—Health Economics Research Center, Faculty of Economics and Business Administration, University of Duisburg-Essen, 45127 Essen, Germany; friederike.arndt@uni-due.de; 2Leibniz Science Campus Ruhr, 45030 Essen, Germany

**Keywords:** biologic, biopharmaceutical, cost control, panel regression, administrative data

## Abstract

Biologic drugs represent a large and growing portion of health expenditures. Increasing the use of biosimilars is a promising option for controlling spending growth in pharmaceutical care. Amid the considerable uncertainty concerning physicians’ decision to prescribe biosimilars, explicit cost control measures may help increase biosimilar use. We analyze the role of regional cost control measures for biosimilars and their association with physician prescriptions in ambulatory care in Germany. We collect data on cost control measures implemented by German physician associations and national claims data on statutory health insurance covering 2009 to 2015. We perform panel regressions that include time and physician fixed effects to identify the average associations between cost control measures and biosimilar share/use while controlling for unobserved physician heterogeneity, patient structure, and socioeconomic factors. We identify 44 measures (priority prescribing, biosimilar quota) for erythropoiesis-stimulating substances, filgrastim, and somatropin. Estimates of cost control measures and their consequences for biosimilar share and use are heterogeneous by drug, measure type, and physician group. Across specialists, biosimilar quotas accounted for 5.13% to 9.75% of the total average biosimilar share of erythropoiesis-stimulating substances. Explicit quota regulations are more effective than priority prescribing. Regional variation in biosimilar use can be partly attributed to the presence of cost control measures.

## 1. Background

Biologic drugs play a decisive role in the treatment of serious diseases and cover a broad range of products [1,2]. Once the patent for a biologic has expired, biosimilars, which imitate approved biologics, may enter the market [3]. Whereas the active ingredients of generic drugs are identical to the chemically synthesized original, biosimilars are only similar to the original biologic. A biosimilar has the same active ingredient as the reference drug, but even small changes in the production process can alter drug efficacy because biopharmaceutical production processes are unique.

Due to the complexity of biologic drug production, price differences between biosimilars and their reference drugs are significantly lower than they are between generic drugs and their identical originals [4,5]. The savings that biosimilar use can provide range from approximately 10% to 35%. Competition in the original biologic and biosimilar drug markets is closer to brand-to-brand competition than to brand-to-generic competition. Physicians are exposed to greater uncertainty when making prescription choices due to the special features that biosimilars have that generic drugs lack [6]. Generally, a higher probability of undesirable side effects, such as immunological reactions, is assumed [7,8]. Therefore, the willingness to use biosimilars depends heavily on the specific product class and condition [9]. Given the smaller price differential and the uncertainty involved, biosimilar diffusion is assumed to be slower than that of generics.

Several policies have been proposed to foster biosimilar use and contain the costs of biologic drug therapies [3,8]. These policies address biologic drug prices (e.g., reference pricing or tendering), reimbursement (e.g., restricting disease indications), or utilization (e.g., physician incentives and market support). The evidence on the effectiveness of tenders in fostering the entry of biosimilar distributors has been mixed [2,3,10]. For example, one additional competitor was associated with a price reduction of 10% in Italy [10]. However, excessive price controls may prevent biosimilar manufacturers from entering markets and lead to drug shortages.

Cost control measures that substitute original biologics for biosimilar drugs have been as widespread as price controls, but their effectiveness has remained untested [8,9]. Cost control measures (e.g., minimum quotas, priority prescribing statutes, substitution policies) can reduce information asymmetries and help save costs. Providing explicit incentives to physicians may be important for increasing biosimilar prescription by influencing physician decision-making and reducing the uncertainty involved in exchanging biologic prescriptions for biosimilars [3]. Studies examining the substitution of brand names by generic drugs have found that physicians’ treatment behavior is highly heterogeneous with regards to their cost-optimal choices [11,12]. Moreover, the spread of information about new prescription options has been shown to be heterogeneous because uncertainties about the treatment effects of new therapy options involve significant learning costs that depend on the physician’s skill and experience [13]. Thus, it is important to account for the heterogeneity of physicians’ treatment behavior. However, no study has yet examined whether cost control measures can increase physicians’ biosimilar use or which measures could produce this result [14].

Finally, biosimilar use has been shown to vary greatly across and within health systems [9]. A growing literature suggests that supply-side economic incentives may be responsible for large differences in health care utilization [15]. Investigating how cost control measures influence biosimilar prescribing can help explain the differences in biosimilar use levels resulting from individual physician prescription decisions conditional on the presence of such measures. Thus, this study identifies the regional variation in the use of cost control measures for biosimilars and the associations between cost control measures and biosimilar prescription by physicians in Germany. As the aim of the biosimilar cost control measures is to increase biosimilar penetration, the objective of this study was to analyze the effects of cost control measures on biosimilar and biologic prescribing at the physician level. We test the hypothesis whether, on average, the prescription of biosimilars in regions that target biosimilar prescription is associated with a higher biosimilar share and utilization compared to regions where such measures are not applied.

## 2. Biosimilars and Cost Control Measures in Germany

The share of biologics (including biosimilars) in Germany’s pharmaceutical market rose steadily to 22.9% (€8.2 billion) in 2015. In that year, 71% of the annual sales of biologics were in the fields of immunology, oncology, and metabolism. In 2016, 240 biologic drug presentations were approved, including biosimilars from seven active substance groups [16]. Germany has one of the world’s highest biosimilar uptake rates [9]. Cost savings were estimated in a counterfactual analysis for the years 2007 to 2014 at many millions of 2006 U.S. dollars (e.g., $258.45 million for epoetin, $143.4 million for filgrastim) whereas the use of biosimilar somatropin leads to additional expenses of $48.74 million [2]. The total cost of biologics was projected to reach €65 billion if no biosimilars had entered the market by 2020 [17].

Physicians who provide services under statutory health insurance are organized at the regional level. Membership of one of Germany’s 17 physician associations (PAs, German: Kassenärztliche Vereinigungen) is mandatory [18]. PAs are responsible for negotiating collective contracts with sickness funds, performing quality assurance, and setting budgets to control the growth of health expenditures (German Social Code Book (§77 SGB V)). Sickness funds and PAs negotiate regional drug agreements on pharmaceutical expenditures (Arzneimittelvereinbarungen) each year to ensure an economic, needs-based, and quality-assured supply [19]. Expenditure control measures, such as recommendations for priority prescriptions and quota setting, are intended to achieve a collaboratively defined expenditure volume. Minimum quotas are formulated as a proportion of the defined daily dose of the preferred drug in a specific drug class. Biosimilar quotas are typically monitored at the physician level, unlike preferred drug lists and closed formularies, for which prior authorization is requested at the patient level [20]. Access is always granted, and no patient groups are excluded once a cost control measure is applied.

Non-compliance with cost control measures for biosimilars, that means the physician’s prescribed share of biosimilars exceeds the quota stated by the PA, may lead to a potential recourse claim of the physician when exceeding the total stated budget. However, physicians are typically not held liable for exceeding the biosimilar quota alone. Enforcement mechanisms, however, may vary across PAs and over time. Some PAs use cost control measures as a guideline (e.g., in Berlin and Hamburg). Other PAs adjust drug budget reviews based on their compliance with drug class-level prescribing measures (e.g., in Brandenburg and Saxony). Drug budget reviews target random draws of physicians and investigate whether the physician’s expenditures exceeded a pre-defined drug budget [21]. In North Rhine, physicians are exempt from a budgetary review if they have fully complied with cost control measures. Due to the high savings potential of biosimilars, PAs began to agree on cost control measures as early as 2008. Biologic and biosimilar distributors entered the German pharmaceutical market uniformly, preventing regional variations in the licensing of drug presentations.

## 3. Methods

### 3.1. Data

We analyze physician prescription behavior in the context of cost control measures by combining data on the biosimilar cost control measures used by German PAs and statutory health insurance claim data at the physician level for each quarter between 2009 and 2015.

For each PA, based on their drug agreements, we captured the type of control mechanism (priority prescribing or explicit biosimilar quota) and the time at which the cost control measure was active. In priority prescribing, biosimilars are prioritized in the physician’s prescription decision, whereas biosimilar quotas provide physicians with a reference value of the biosimilar share. The drug agreements were identified through a structured search of PAs’ websites and contact with physician associations where data was not available from the web (Appendix A). Three of the 17 PA regions (Hesse, Brandenburg, and Mecklenburg-Western Pomerania) were excluded, as the drug agreements collected did not cover the entire study period.

Our main source was ambulatory care prescription data (Arzneiverordnungsdaten according to § 300 Abs. 2 SGB V, provided by Zentralinstitut für die Kassenärztliche Versorgung in Deutschland, Zi), which include all prescriptions that have been filed by the patients of statutory health insurance in a physician’s practice. The data provide information on prescribed drugs (identification code, number of prescriptions, price, and dispensation date), the prescribing physician, the region, and the specialist group.

The outpatient physician dataset (Ambulante vertragsärztliche Abrechnungsdaten according to § 295 SGB V, provided by Zi) contains information on patients who had at least one physician contact during the reporting period. We captured the age, gender, and average morbidity structure of biologic patients in each practice. Morbidity was calculated as a relative risk score based on the diagnosis-based patient classification system to capture the morbidity used as a basis for the settlement of ambulatory reimbursement rates at the regional level [22].

Data on average incomes in the physicians’ practice area and urbanization status were retrieved from the Federal Institute for Research on Building, Urban Affairs and Spatial Development (INKAR-Daten, https://www.inkar.de/).

### 3.2. Outcome Variables

We analyzed the associations between cost control measures and physician prescription by capturing several outcome variables. Most importantly, the extent of target achievement is reflected in the biosimilar share, defined as the proportion of prescribed biosimilars in all biologics prescribed at a given time for a biologic drug. We also captured use levels as the number of biosimilar prescriptions, biologic prescriptions, and original biologic prescriptions per 1000 patients in a physician’s practice.

### 3.3. Variables Accounting for Heterogeneity of Prescribing Behavior

We accounted for heterogeneity in physician prescription across practices by capturing the characteristics of the prescribing physician, patient composition with regards to biologic patients, and the socio-economic factors in the area where the physician’s practice was located. For the characteristics of the physician’s practice, we accounted for whether the physician was practicing in a group practice or solo practice, as well as the number of patients in the practice. To account for the characteristics of patients receiving a biologic drug, we captured the percentage of patients receiving biologics out of all patients in the practice, the proportion of patients over 65 years of age, the proportion of male patients, and the average morbidity score of these patients. As physicians may vary in their price sensitivity in the prescription of biologic drugs, we captured the average gross sales price per defined daily dose of the biologics the physician was prescribing [23].

Regarding regional characteristics, we captured the average household income in the region in which the physician’s practice was located. We calculated mean values for the 88 regional units to which a physician’s practice was assigned by PAs, weighted by the number of inhabitants in that regional unit. Finally, we captured the district type, classified as agglomeration, urbanized, or rural.

### 3.4. Statistical Analysis

We performed fixed effects panel regressions for biologic drugs and according to physician specialist status (i.e., general practitioners (GPs) and specialists) to analyze the effect of cost control measures on biosimilar share and utilization. Physicians were considered if they prescribed three or more prescriptions of the biologic in a quarter (three is the minimum number of observations possible per unit to observe when analyzing data according to § 300 Abs. 2 SGB V). We estimated separate regressions accounting for (a) the full sample of physicians and (b) physicians prescribing in each quarter of the observation period (i.e., regular prescribers). When we considered physicians who prescribed consistently (i.e., in the balanced panel), the number of observed physician–quarter combinations decreases notably (e.g., for erythropoiesis-stimulating substances (ESAs) from 50.389 to 23.258 for specialists and from 15.506 to 1.074 for GPs). Thus, in our fixed effects approach, the number of observations for GPs who prescribed filgrastim and somatropin regularly was not always sufficient; we thus report only the results where the F-statistic was significant.

First, we identified associations between PA cost control measures and the outcome variables, accounting for practice characteristics, patient structure, and regional socio-economic factors. We performed the following panel regressions that identify the effect of cost control measures on outcomes related to biosimilar share and biosimilar and biologic use, which are the primary targets of the policy:(1)$$Y_{ijt}=\beta_{0}+\beta_{1}Priority_{it}+\beta_{2}Quota_{it}+\beta_{3}X_{ijt}+\theta_{i}+v_{t}+u_{ijt}$$

The dependent variable $Y_{ijt}$ reflects any of the outcome variables at the level of physician j in PA i at time t. Priority=1 if, in PA i in quarter t, there was a specification for priority prescribing, and 0 otherwise. Thus, the coefficient $\beta_{1}$ indicates the average effect of a priority prescribing policy on biosimilar share and use across all PAs with that policy compared to PAs without such a policy. Quota=1 was defined if a biosimilar quota was specified in PA i in quarter t and 0 otherwise. Thus, the coefficient $\beta_{2}$ indicates the average effect of a biosimilar quota policy on biosimilar share and use across PAs with that policy compared to PAs without such a policy. The explanatory variables are represented by the vector $X_{ijt}$. These describe the characteristics of the prescribing physician, patient composition with regards to biologic patients, and the socio-economic factors of the area in which the physician’s practice was located. $\theta_{i}$ considers PA fixed effects to account for level differences across regions. $v_{t}$ reflects time-specific fixed effects to consider general time trends in outcome variables. $u_{ijt}$ specifies the error term. Robust error terms have been generated to avoid heteroskedasticity. We fitted the panel models using the xtreg procedure in STATA 15 (College Station, TX, USA).

As the literature suggests that there is substantial unobserved heterogeneity in physician behavior, which goes beyond the observable characteristics that we could capture in the ambulatory claims data [24,25], we additionally estimated two-way fixed effects models using time ($v_{t}$) and physician ($\gamma_{j}$) fixed effects. In contrast to our first specification, we refrain from control variables that vary by physician, as such variation will be captured in the physician fixed effect. The interpretation of the variables $\beta_{1}$ and $\beta_{2}$ is analogous to the specification in Equation (1). We also continued to control for regional-level socioeconomic factors $(X_{ijt}^{s}$):(2)$$y_{ijt}=\beta_{1}Priority_{it}+\beta_{2}Quota_{it}+\gamma_{j}+v_{t}+u_{ijt}$$

### 3.5. Ethics Approval

Ethical approval was not requested for the research performed for this manuscript. The study used anonymized secondary data that comply with the data protection standards outlined in the German Social Code Book.

## 4. Results

### 4.1. Descriptives

We identified 44 cost control measures for three of the seven biologic drugs for which biosimilars were available (see Table 1 and Appendix A): erythropoiesis-stimulating substances (ESAs), filgrastim, and somatropin [16]. Drug agreements also existed for biosimilar versions of infliximab, which we excluded from further analysis because cost control measures by PAs were introduced during the biosimilar market entry of infliximab in 2015.

The variation in biosimilar use is wide across drugs and PAs. Figure 1 shows biosimilar shares in 2009. For ESAs, the difference between the PA with the lowest (Saarland, 2.07%) and highest biosimilar share (Bremen, 35.95%) was 33.83 percentage points with a mean share of 18.06%. By 2015, the biosimilar share increased to a mean value of 37.74% across PAs. However, the range of difference in biosimilar shares across PAs was about equal between 2015 and 2009, with the lowest in Baden-Wuerttemberg (17.45%) and the highest in North Rhine (50.04%). For filgrastim, the average change in biosimilar share was highest and increased by 48.63 percentage points from 2009 to 2015. Again, the difference in biosimilar share between the lowest and highest PA was considerable but decreased from 46.84% to 16.83% by 2015. For somatropin, the mean biosimilar share across PAs was the lowest across the biologic drugs considered (13.79% in 2009), and the average change in biosimilar share was +10.21 percentage points (2009‒2015).

Biosimilar shares and use rates differ widely according to the physician’s specialization status (see Table 2). Overall, the proportion of patients treated in a practice with biologic drugs for which cost control measures were implemented was low, and the share of patients receiving a biologic drug was four to 16 times higher in specialist practices than among GPs.

Table 3 reports descriptive statistics for the control variables of all physicians prescribing biologics in 2013 at the physician–practice level. The patient characteristics vary considerably within and across the biologic drugs. Approximately 80% of the observed physician–quarter combinations relate to specialists’ prescriptions.

### 4.2. Associations between Cost Control Measures and Biosimilar Use

The panel regression results suggest that the associations between cost control measures and the biosimilar share differ by the type of measure, drug, and physician specialty, as well as whether all physicians prescribing biologic drugs are considered or only those prescribing biologic drugs regularly (see Table 4). Where we identified significant estimates, our estimates account for 8.1% to 44.77% of the average biosimilar share for the corresponding physician groups for 2009 to 2015 (see Table 5).

The panel regression results reveal negative associations with the biosimilar share across all three substances when a priority prescription cost control measure was present in a PA. However, this estimate is significant only for specialists prescribing filgrastim (−3.31 percentage points on average, *p* < 0.01). By contrast, our results suggest that, on average, when explicit biosimilar quotas were active, the biosimilar share was significantly higher for regular prescribers of biologic drugs but not for all prescribers. For ESAs, the presence of a biosimilar quota was associated with a significantly increased share for regularly prescribing GPs (+9.24 percentage points, *p* < 0.1) and specialists (+1.78 percentage points, *p* < 0.05). For filgrastim, the presence of a biosimilar quota was significantly associated with an average increase in biosimilar share across all specialists (+6.12 percentage points, *p* < 0.05) compared to absence of any cost control measure. When we considered all prescribers, the regression analyses revealed that, for ESAs, the presence of a biosimilar quota was associated with a small but significant decrease in the biosimilar share among GPs and specialists (−2.6 and −0.98 percentage points, respectively, *p* < 0.1). For somatropin, the presence of a cost control measure was not significantly associated with the biosimilar share in any of the models estimated.

In considering the control variables capturing the characteristics of the physician’s practice, biologic patient composition and regional level socio-economic factors demonstrated that the associations with the presence of a cost control measure had similar magnitudes. Group practice status was most strongly associated with an increased biosimilar share. Practicing in a group practice was associated with a decreased biosimilar share only for GPs prescribing ESAs in the balanced panel and GPs prescribing somatropin in the unbalanced panel. We also identified a positive association between the number of patients, as well as the number of biologic patients in the practice and the biosimilar share. A higher comorbidity index in the physician’s practice was associated with a lower biosimilar share for somatropin. The share of male (compared to female) biologic patients was, when significant, negatively associated with the biosimilar share. The share of patients older than 65 years of age was significantly positively associated with the biosimilar share for physicians prescribing ESAs. Physicians who prescribed more expensive biologics had, on average, a lower biosimilar share.

Regarding regional socio-economic characteristics, the associations between the biosimilar share and the average household income in the region were mostly positive. A negative association was observed only for GPs prescribing ESAs in each quarter and specialists prescribing somatropin. Biosimilar shares were also associated with practice location; the association with rural and urban (compared to metropolitan) regions was mainly positive, except for specialists prescribing somatropin, for whom there was a negative association.

Regarding unobservable heterogeneity—most importantly, physician prescription behavior beyond practice characteristics—the results reported in Table 5 suggest that, for ESAs, the estimates for the presence of a biosimilar quota are higher than in the panel regressions where we do not account for physician-level fixed effects. For all specialists, the effect estimates amount to +2.47 percentage points compared to −0.98 percentage points (see Table 4); for specialists prescribing regularly, the estimate increases to +3.39 percentage points compared to +1.78 (see Table 4). Comparing these effects to the average biosimilar share, these increases account for 7.11–9.76% of the average biosimilar share. Moreover, the presence of a priority prescribing policy was shown to be associated with a significant decrease in biosimilar share (−2.3 percentage points, *p* < 0.1). For both filgrastim and somatropin, we find no significant effect estimates when we account for unobserved physician heterogeneity in the two-way fixed effects regressions.

Table 5 also reports the estimates for the association between cost control measures and biosimilar and biologic drug use. For ESAs, the results show that the increase in biosimilar share when a biosimilar quota was in place was achieved largely due to increases in biosimilar prescriptions (+8.55 (all specialists), +11.5 (balanced panel), *p* < 0.05), while original biologic prescriptions remained unchanged and the total number of prescriptions significantly increased. For the priority prescribing of ESAs, the total increase in biologic prescriptions outpaced the significant increase in biosimilar prescriptions for all specialists and regular prescribers. Contrariwise, while priority prescribing had no effect on regular specialist prescribers, the regression results suggest that, across all specialists, usage volumes decreased significantly (−6.89 for biosimilar prescriptions, *p* < 0.1; −11.01 for total biologic prescriptions, *p* < 0.1). This indicates that biosimilars were only partly used to substitute for original biologics but this also led to an increase in the usage of these agents.

For filgrastim, unlike in the panel analysis results, the negative association between priority prescribing and the biosimilar share was not significant once we controlled for unobserved physician heterogeneity. Nevertheless, the additional utilization outcomes reveal that the presence of priority prescribing was significantly associated with an increased prescription of biosimilars by specialists (+3.33 prescriptions, *p* < 0.1) and a similar increase in the number of all biologics prescribed by specialists (+3.98 prescriptions, *p* < 0.1). Again, we find no significant associations between biosimilar share and usage measures for somatropin. The regression results were robust in terms of the effects’ magnitude and significance, independent of whether we specified biosimilar share and usage outcomes by the number of prescriptions or defined daily doses.

## 5. Discussion

This study finds regional variation in the use of cost control measures (priority prescribing, biosimilar quota) for biosimilars across PAs aiming to control prescription drug costs in Germany. The objective of our study was to analyze the associations of cost control measures and biosimilar and biologic prescribing at the physician level. We show that, for specialists, average changes due to the presence of cost control measures account for 5.13% to 9.75% of the average biosimilar share of ESAs from 2009 to 2015. Given that the sanctioning of physicians for non-compliance with biosimilar cost control measures is very limited, these associations are substantial in markets where annual growth in the biosimilar share has been estimated at 2.6 to 9 percentage points [2]. Thus, we cannot reject our hypothesis that cost control measures are associated with physician prescribing of biologics and biosimilars. We also demonstrate that, while the average effects of cost control measures may be negative across all physicians, cost control measures seem to be especially effective in the regular prescriber group—specialists in our case. Thus, cost control measures are shown to change prescribing behaviors for certain biologic drugs and prescribers on average across all regions and measures.

Our findings conflict with cross-country comparisons of biosimilar penetration, which have shown that the presence of biosimilar quotas does not increase biosimilar share [2]. Similar to other studies on regional variation of health care provision, this finding may be explained by the fact that many cost control measures are implemented at a lower than national level. Variation in biosimilar prescribing within one region may also be at least as large, or larger, as variation across regions [1], as for the biosimilar cost control measures studied here. If within-country variation is large and regions respond heterogeneously, this may cancel out effects on a national level. Thus, the implementation of such measures may vary strongly across regions, leading to large heterogeneity across biosimilar penetration levels and heterogeneous effects within Germany’s health system.

As uncertainty regarding whether biosimilars are equivalent to their original biologic companions is substantial, information disclosure by authorities, such as PAs, is vital for encouraging an uptake in biosimilar drugs, especially among regular prescribers. As with the substitution of generics for brand name drugs, physicians are important agents in patients’ decisions to use biosimilars or original biologics. Here, state-level substitution policies in countries outside Germany have not always been shown to be effective [11]. Our study highlights that when comparing cost control measures chosen by PAs an explicit quota was more effective in fostering ESA and filgrastim biosimilar use than priority prescribing.

Cost control measures’ capacity to increase the biosimilar share depends on the specific biologic drug, the physician’s specialization, and whether physicians prescribe biosimilars on a regular basis. Previous studies have also shown that the market profiles of biologic drugs are highly heterogeneous, as is reflected in our results on the effect sizes of the associations, with ESAs being the highest and somatropin the lowest [3,26]. As cost control measures primarily target specialists, associations with the biosimilar share are generally higher for them than for GPs. The regression results also reveal that, besides selected practice characteristics, biologic patient base, and regional factors, physician heterogeneity also relates to the idiosyncratic tastes and preferences of the physicians. If these are not accounted for, the identified associations with cost control measures may be biased.

Priority prescribing may be considered a weaker method of incentivizing physicians to prescribe more biosimilars, as it showed partly negative associations with the average biosimilar share. Whereas the measure was negatively associated with the use of ESA biosimilars and total ESA biologic prescriptions, it was positively associated with the use of filgrastim biosimilars by specialists, although the total increase in total biologic filgrastim was larger. Thus, we find that the presence of a cost control measure may also affect the total number of biologic prescriptions. This type of pattern is in stark contrast to how physicians in Germany react to other cost control measures, such as drug budgets, where use levels have been shown to remain unchanged [21].

PA-level cost control measures for somatropin were likely not effective due to the higher uncertainty regarding the difference in efficacy between the original biologic and the biosimilar [20] and the competitive structure. In 2020, only one biosimilar (omnitrope) has been approved, while five to seven presentations are available for ESAs and filgrastim.

This study has several limitations. Besides the hidden discount agreements between manufacturers and sickness funds, we cannot account for regional variation in promotional activity. In addition, we were unable to identify variations in other activities at the PA level, such as quality circles and circulars, as well as additional prescribing restrictions at the practice level, such as drug budgets [21]. Moreover, we could not stratify the results by patients initiating biologic drug treatment, for which it may be easier to start biosimilar treatment instead of switching drugs. Since the analyses are based exclusively on claims data from ambulatory care, strategic reactions like the referral behavior of physicians between practices and hospitals could not be observed. Finally, our results identify the average effects of the cost control measures across all PAs, although the effectiveness of the policies may vary within the different regions. Future analyses could therefore perform separate analyses by PA in a quasi-experimental design.

This study has several implications for health policy and prescribing physicians. Although the average effects across all physicians and specialists may be small, changes in biosimilar usage may be largely driven by a small group of specialist regular prescribers. Supporting this group of physicians with information that reduces the uncertainty involved in switching from biologics to biosimilars could strengthen the effectiveness of cost control measures and thus increase the savings potential. Activities for accomplishing this could include targeted quality circles and small peer groups seeking to improve the standard practice [27].

The savings generated by cost control measures will be meaningful, given the identified associations between cost control measures and biosimilar use for at least one drug class (ESAs), the fact that biosimilars are generally 10% to 35% cheaper, and the fact that biosimilar prices were decreasing in the observation period [26]. As information on discounts is typically not disclosed, we cannot identify the exact net savings to statutory health insurance. Moreover, manufacturers may respond strategically to the introduction of internal reference pricing as for ESAs in October 2012 [28].

## 6. Conclusions

Biologic drugs are a large and growing portion of health expenditures. This study examines the role of regional biosimilar cost control measures for three biologic drugs in Germany. We show that cost control measures for biosimilars contributed to increasing the biosimilar share across regions that made use of such policies. However, the type of cost control measure (priority prescribing vs. quota) matters, and average effects are not uniform across the drugs studied and physician groups targeted. Biosimilar quotas may be a vital tool for increasing the biosimilar share by reducing uncertainty among physicians who prescribe biosimilars regularly, but it may also increase overall biologic drug use.

## Figures and Tables

**Figure 1 ijerph-17-04113-f001:**
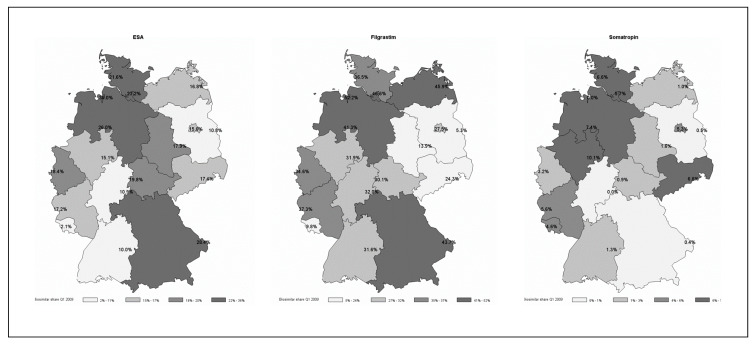
Biosimilar share of biologic drugs exposed to cost control measures by physician associations, first quarter (Q1) 2009.

**Table 1 ijerph-17-04113-t001:** Biologic drugs with availability of biosimilars and cost control measures by physician associations, 2009–2015.

Drug (Class)	Disease Areas	Prescriptions (2009–2015)	Biosimilar Quota	Priority Prescribing	Combi-Nation	Total
Erythropoesis stimulating agents	Anemia, cancer, postoperative inflammation, reactions after kidney transplantation	3,784,943	14	4	3	15
Filgrastim	Neutropenia, severe neutropenia in cancer, cancer	330,081	3	8	2	9
Somatropin	Growth disorders	202,519	6	9	5	10
	Total	4,317,543	23	21	10	44

**Table 2 ijerph-17-04113-t002:** Descriptive statistics of biologic and biosimilar prescribing by biologic drug, aggregated at PA level, 2009–2015.

Drug (Class)	Specialization	Statistic	Biosimilar Share (Prescriptions)	Biosimilar Share (DDD ^2^)	Share of Patients Using Biosimilars in Practice	Number of Patients Using Biosimilars in Practice	Prescriptions Biologics	Prescription Biosimilars	Prescriptions per 1000 Patients, Biologics	Prescriptions per 1000 Patients, Biosimilars
ESAs ^1^	GP	mean	20.64%	20.49%	0.31%	2.53	10.9	2.99	429	111
		(s.d.)	(6.51%)	(6.38%)	(0.33%)	(2.4)	(10.1)	(2.95)	(419)	(97.3)
	Specialist	mean	34.73%	31.92%	3.37%	23.8	66.3	28.4	2801	947
		(s.d.)	(9.24%)	(8.7%)	(1.14%)	(6.43)	(11.1)	(7.22)	(547)	(273)
Filgrastim	GP	mean	61.85%	61.85%	0.33%	4.36	7.37	5.35	28.7	20.1
		(s.d.)	(15.86%)	(15.97%)	(0.27%)	(3.03)	(3.34)	(3.35)	(15.6)	(14.4)
	Specialist	mean	75.94%	75.93%	1.29%	7.5	11.4	9.06	110	79.6
		(s.d.)	(6.82%)	(6.78%)	(0.63%)	(2.11)	(2.67)	(2.59)	(55.7)	(43.8)
Somatropin	GP	mean	3.20%	3.14%	0.04%	0.33	6.27	0.34	409	26.5
		(s.d.)	(3.99%)	(3.99%)	(0.08%)	(0.42)	(2.5)	(0.42)	(659)	(56.7)
	Specialist	mean	9.51%	10.33%	0.64%	2.46	23.5	2.51	4456	626
		(s.d.)	(4.45%)	(5.12%)	(0.37%)	(1.23)	(7.05)	(1.25)	(2695)	(424)

^1^ Erythropoesis-stimulating agents; ^2^ defined daily dose; ^3^ general practitioner.

**Table 3 ijerph-17-04113-t003:** Descriptive statistics of control variables and variables to stratify physicians by active ingredient, 2013.

Drug (n = Number of Physician–Quarter Combinations)	Statistic	Erythropoesis-Stimulating Agents (n = 10,948)		Filgrastim (n = 1516)		Somatropin (n = 3193)		Total (n = 19,707)	
Number of patients in practice receiving biologic drug	mean	54.17		19.25		23.47		37.35	
	(s.d.)	(57.32)		(23.16)		(42.08)		(49.51)	
Share of patients receiving biologic drug in practice	mean	0.03		0.01		0.01		0.02	
	(s.d.)	(0.07)		(0.03)		(0.02)		(0.05)	
Patient age (biologic patients)	mean	70.45		59.53		25.42		59.77	
	(s.d.)	(8.35)		(10.94)		(18.79)		(17.63)	
Comorbidity index of biologic patients	mean	17.26		13.54		9.06		14.54	
	(s.d.)	(3.60)		(3.14)		(3.85)		(4.75)	
Share of male individuals of biologic patients	mean	0.50		0.39		0.57		0.49	
	(s.d.)	(0.23)		(0.28)		(0.26)		(0.25)	
Share of individuals older than 65 years of age of biologic patients	mean	0.74		0.47		0.07		0.51	
	(s.d.)	(0.23)		(0.25)		(0.19)		(0.36)	
Total number of patients in practice	mean	1362.51		1704.44		1939.04		1584.65	
	(s.d.)	(1161.10)		(1493.27)		(2202.65)		(1449.43)	
Price per defined daily dose of packages prescribed by physician (biologics incl. biosimilars)	mean	10.98		175.21		53.24		45.84	
	(s.d.)	(3.13)		(94.88)		(18.22)		(70.07)	
Average household income in region of practice in 1000 EUR	mean	1699.60		1695.37		1702.37		1702.10	
	(s.d.)	(173.48)		(177.99)		(167.78)		(173.82)	
		N	%	N	%	N	%	N	%
Type of practice									
Solo practice		4206	38.42	1059	69.85	1748	54.74	9360	47.5
Group practice		6742	61.58	457	30.15	1445	45.26	10,347	52.5
Area									
Metropolitan		5409	49.41	814	53.69	1637	51.27	9864	50.05
Urbanized		4129	37.71	575	37.93	1165	36.49	7343	37.26
Rural		1410	12.88	127	8.38	391	12.25	2500	12.69
Specialist status									
General practitioner		2470	22.56	293	19.33	482	15.1	3637	18.46
Specialist		8478	77.44	1223	80.67	2711	84.9	16,070	81.54

**Table 4 ijerph-17-04113-t004:** Panel regressions of biosimilar share by drug (class), all physicians, and balanced panel, 2009–2015.

	ESA				Filgrastim			Somatropin		
	All Physicians		Balanced Panel		All Physicians		Balanced Panel	All Physicians		Balanced Panel
	GP	specialist	GP	specialist	GP	specialist	specialist	GP	specialist	specialist
Priority prescribing	−0.0263	−0.0069	−0.0776	−0.0129	−0.0104	−0.0331 ***	−0.0180	0.0095	−0.0077	−0.0199
	(0.0202)	(0.0083)	(0.0665)	(0.0106)	(0.0270)	(0.0097)	(0.0230)	(0.0157)	(0.0069)	(0.0105)
Quota	−0.0260 *	−0.0098 *	0.0924 *	0.0178 **	−0.0356	0.0612 **	0.0411	−0.0087	0.0152	0.0118
	(0.0105)	(0.0046)	(0.0359)	(0.0056)	(0.0735)	(0.0220)	(0.0589)	(0.0141)	(0.0083)	(0.0138)
Group practice (ref: solo practice)	0.0153 *	0.0502 ***	−0.0434 *	0.0241 ***	0.1081 ***	0.0967 ***	0.0695 ***	−0.0250 **	−0.0053	−0.0405 ***
	(0.0069)	(0.0031)	(0.0181)	(0.0041)	(0.0159)	(0.0062)	(0.0149)	(0.0093)	(0.0056)	(0.0091)
Share of patients receiving biologic drug in practice	2.6378 ***	1.9431 ***	3.9994 ***	2.2698 ***	8.4402 ***	1.8534 ***	7.1655 ***	14.0649	2.9614 ***	3.8537 ***
	(0.2882)	(0.0390)	(0.3167)	(0.0480)	(1.7605)	(0.1609)	(1.5168)	(8.3375)	(0.4713)	(0.4650)
Patient age	0.0021 ***	0.0020 ***	−0.0032	−0.0009	0.0064 ***	0.0041 ***	−0.0044	−0.0004	−0.0010 ***	0.0002
	(0.0004)	(0.0003)	(0.0033)	(0.0006)	(0.0010)	(0.0008)	(0.0025)	(0.0004)	(0.0002)	(0.0006)
Comorbidity index	−0.0001	0.0017 **	0.0017	−0.0011	0.0041	0.0062 ***	−0.0014	−0.0015 *	−0.0023 **	−0.0054 *
	(0.0006)	(0.0006)	(0.0024)	(0.0009)	(0.0022)	(0.0012)	(0.0041)	(0.0007)	(0.0009)	(0.0023)
Share male individuals of biologic patients	−0.0068	−0.0207 *	0.0117	−0.0500 *	−0.0193	−0.1019 ***	−0.2175 ***	−0.0087	−0.0148	−0.0483
	(0.0082)	(0.0091)	(0.0947)	(0.0213)	(0.0251)	(0.0126)	(0.0443)	(0.0096)	(0.0094)	(0.0272)
Share individuals older 65 years of age of biologic patients	−0.0144	0.0383 ***	0.4178 ***	0.1494 ***	−0.0114	−0.0127	0.0417	−0.0156	0.0149	−0.0479
	(0.0114)	(0.0116)	(0.1072)	(0.0233)	(0.0354)	(0.0211)	(0.0760)	(0.0145)	(0.0172)	(0.0437)
Number of patients in practice in 1000 patients	0.0054	0.0182 ***	0.0445 ***	0.0478 ***	−0.0091	0.0154 ***	0.0647 ***	0.0157 ***	0.0051 **	0.0099 ***
	(0.0033)	(0.0015)	(0.0105)	(0.0027)	(0.0063)	(0.0023)	(0.0116)	(0.0031)	(0.0016)	(0.0025)
Price per DDD of packages prescribed by physician	−0.0014	−0.0003 ***	−0.0155 ***	−0.0003	−0.0006 ***	−0.0008 ***	−0.0020 ***	−0.0003 *	−0.0008 ***	−0.0013 ***
	(0.0011)	(0.0001)	(0.0037)	(0.0003)	(0.0001)	(0.0001)	(0.0002)	(0.0001)	(0.0001)	(0.0001)
Average household income in region of practice in 1000 EUR	−0.0146	0.1053 ***	−0.4648 ***	0.1230 ***	0.2550 **	0.3196 ***	0.1587	0.1009 **	−0.1499 ***	−0.2061 ***
	(0.0384)	(0.0156)	(0.1164)	(0.0204)	(0.0867)	(0.0318)	(0.0812)	(0.0343)	(0.0252)	(0.0416)
Urbanized areas (ref: metropolitan)	−0.0056	0.0351 ***	0.0624 **	0.0440 ***	−0.1306 ***	−0.0087	0.0540 *	0.0283 *	0.0036	−0.0434 ***
	(0.0078)	(0.0035)	(0.0207)	(0.0045)	(0.0197)	(0.0071)	(0.0225)	(0.0110)	(0.0053)	(0.0092)
Rural (ref: metropolitan)	−0.0233	0.0795 ***	0.0328	0.0777 ***	−0.1085 **	0.0229 *	0.1099 ***	0.0128	−0.0633 ***	−0.0970 ***
	(0.0127)	(0.0055)	(0.0480)	(0.0069)	(0.0340)	(0.0113)	(0.0254)	(0.0132)	(0.0078)	(0.0100)
TIME Fixed Effects	✓	✓	✓	✓	✓	✓	✓	✓	✓	✓
Physician Association Fixed Effects	✓	✓	✓	✓	✓	✓	✓	✓	✓	✓
Intercept	0.1310	−0.1684 ***	0.7707 **	−0.0614	−0.3750 *	−0.2740 **	0.8661 ***	−0.1694 **	0.3906 ***	0.7799 ***
	(0.0692)	(0.0305)	(0.2545)	(0.0448)	(0.1622)	(0.0850)	(0.2267)	(0.0639)	(0.0459)	(0.0753)
N	15,506	50,389	1074	23,258	3021	14,758	1607	1.436	7392	2569
r2	0.070	0.319	0.537	0.456	0.308	0.247	0.504	0.186	0.216	0.376
F	13.339	313.778	58.227	312.700	25.908	66.053	93.470	3.877	27.386	23.007
RMSE	0.3568	0.2857	0.2041	0.2396	0.3768	0.3307	0.2594	0.1305	0.1675	0.1481

ESA: Erythropoesis stimulating agents; GP: general practitioner; ***: *p* < 0.01; **: *p* < 0.05; *: *p* < 0.1, standard errors are in parantheses; The balanced panel includes physicians prescribing the target biologic drug in each quarter 2009–2015.

**Table 5 ijerph-17-04113-t005:** Two-way fixed effects regression by drug (class), specialization, and outcome; associations with PA measures for biosimilar prescribing, controlling for physician and time fixed effects, all physicians, and balanced panel, 2009–2015.

Biologic Drug (Class)	Outcome	Biosimilar Share (Prescriptions)		Prescriptions Biosimilars		Prescriptions Original		Prescriptions Total	
	Type of Measure	GP	Specialist	GP	Specialist	GP	Specialist	GP	Specialist
ESAs (n = 15,506 (GPs), n = 50,389 (specialists))	Priority prescribing	−0.0161	−0.0230 *	−1.7419 *	−6.8939 *	−0.6267	−4.1181	−2.3686	−11.0120 *
		(0.0180)	(0.0106)	(0.8732)	(3.4506)	(1.3009)	(3.2010)	(1.7723)	(4.9739)
	Quota	−0.0044	0.0247 ***	−0.5899	8.5505 **	−1.3458	2.2037	−1.9357	10.7542 *
		(0.0148)	(0.0075)	(1.4571)	(2.7040)	(1.9901)	(2.7980)	(2.9063)	(4.3682)
Filgrastim (n = 3021 (GPs), n = 14,758 (specialists))	Priority prescribing	−0.0196	−0.0055	0.2571	3.3308 *	0.903	0.6469	1.1601	3.9777 *
		(0.0361)	(0.0126)	(0.8610)	(1.6303)	(0.6409)	(1.2059)	(0.9966)	(1.9138)
	Quota	−0.1228	0.0442	2.1511	−1.3236	8.313	−1.1173	10.464	−2.4409
		(0.1073)	(0.0319)	(1.1118)	(3.5723)	(6.9305)	(1.8316)	(7.2926)	(4.4568)
Somatropin (n = 1436 (GPs), n = 7392 (specialists))	Priority prescribing	0.0055	−0.0018	0.0834	0.069	0.7486	−10.5695	0.832	−10.5005
		(0.0231)	(0.0087)	(0.1211)	(1.8285)	(1.3554)	(11.9035)	(1.4192)	(13.1738)
	Quota	−0.0224	0.0136	0.0997	0.4783	1.6937	−12.3819	1.7933	−11.9036
		(0.0340)	(0.0106)	(0.1958)	(1.6254)	(2.3429)	(11.4075)	(2.490)	(12.3867)

ESAs: Erythropoesis-stimulating agents; GP: general practitioner; ***: *p* < 0.01; **: *p* < 0.05; *: *p* < 0.1, standard errors are in parentheses.

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
