# Peer review of "Physician-Level Cost Control Measures and Regional Variation of Biosimilar Utilization in Germany"

_ijerph, 2020, doi:10.3390/ijerph17114113_

Round 1
Reviewer 1 Report
Thank you, I really enjoyed reading your interesting and well-written manuscript. I have made some comments below.
Abstract: The abstract is clear and reflects the paper content well.
- Background
The background is well presented and sets the research in clear context. The description of biosimilars from financial and clinical perspectives is appropriate, as is the contextual information re the German system. I have very few comments.
Line 47-48: Strategies and policies are slightly different. If you mean policy, it would be better to be consistent with the terminology here.
- Biosimilars
Lines 78-79: It is not quite clear whether you are referring to biosimilars or biologics in this part. Maybe consider removing the opening sentence and putting it after the general discussion of biologics for clarity.
Lines 98-99: I am not sure what ‘non-compliance with cost-control measures has indirect…..’ means. Please revise for clarity.
- Methods
Line 120-121: Could you clarify if these databases have complete data (e.g. is all/a proportion of the prescribing captured in the prescription data, is this the only place all of the prescription data is found etc.?)
Lines 158-159: Why was three selected as the minimum for physician inclusion?
- Results
Table 1: What is the purpose of including the ICD classifications?
- Discussion
Lines 310-313: How do you explain this difference, could there be reasons why?
Reviewer 2 Report
“Physician-level cost-control measures and regional variation of biosimilar utilization in Germany”
The article idea is very good, suitable for the current context. The author focuses on analyzing data from 2005 to 2019 with quite a large number of survey samples. The article is very relevant to the health major this time especially when Germany is facing an outbreak. Only Identifies the regional variation in the use of cost-control measures for biosimilars and the associations between cost-control measures and biosimilar prescriptions by physicians in various regions in Germany.
However, I have some suggestions for the author as follows:
- What is the research objective? How to control costs?
- Please explain how the questionnaire and the design model are developed, how to make sure this model to fit with this data set?
- Hypotheses of this research haven’t demonstrated.
- Conclusions and discussion make the reader confused readers because the author did not specify how the goals were addressed in the conclusions.
Reviewer 3 Report
The authors presented this study to identify the regional variation in the use of cost-control measures for biosimilar and the associations between cost-control measures and biosimilar prescription by physicians in Germany. The paper is well organized and well presented. The experiments and results are well described with real data collected from 2009 to 2015. The paper can be accepted after minor revision: (1) There are some formats need to be edited, such as figure captions, and supplementary tables. (2) The panel model parameter are recommended to be introduced more clearly. (3) If comparison experiments is conducted, and it's better. Overall, it is a good manuscript for the cost-control measures for biosimilar, which uses suitable measures and abundant data, and some valuable results have been obtained.Author Response
Please see the attachment.
